# CRB2 Loss in Rod Photoreceptors Is Associated with Progressive Loss of Retinal Contrast Sensitivity

**DOI:** 10.3390/ijms20174069

**Published:** 2019-08-21

**Authors:** C. Henrique Alves, Nanda Boon, Aat A. Mulder, Abraham J. Koster, Carolina R. Jost, Jan Wijnholds

**Affiliations:** 1Department of Ophthalmology, Leiden University Medical Center (LUMC), Albinusdreef 2, 2333 ZA Leiden, The Netherlands; 2Department of Cell & Chemical Biology, Leiden University Medical Center (LUMC), 2300 RC Leiden, The Netherlands; 3Netherlands Institute for Neuroscience, an Institute of the Royal Netherlands Academy of Arts and Sciences (KNAW), Meibergdreef 47, 1105 BA Amsterdam, The Netherlands

**Keywords:** crumbs complex, crumbs homolog-1, Leber congenital amaurosis, photoreceptors, retina, retinal degeneration, retinitis pigmentosa, rod photoreceptors

## Abstract

Variations in the Crumbs homolog-1 (*CRB1*) gene are associated with a wide variety of autosomal recessive retinal dystrophies, including early onset retinitis pigmentosa (RP) and Leber congenital amaurosis (LCA). CRB1 belongs to the Crumbs family, which in mammals includes CRB2 and CRB3. Here, we studied the specific roles of CRB2 in rod photoreceptor cells and whether ablation of CRB2 in rods exacerbates the *Crb1*-disease. Therefore, we assessed the morphological, retinal, and visual functional consequences of specific ablation of CRB2 from rods with or without concomitant loss of CRB1. Our data demonstrated that loss of CRB2 in mature rods resulted in RP. The retina showed gliosis and disruption of the subapical region and adherens junctions at the outer limiting membrane. Rods were lost at the peripheral and central superior retina, while gross retinal lamination was preserved. Rod function as measured by electroretinography was impaired in adult mice. Additional loss of CRB1 exacerbated the retinal phenotype leading to an early reduction of the dark-adapted rod photoreceptor a-wave and reduced contrast sensitivity from 3-months-of-age, as measured by optokinetic tracking reflex (OKT) behavior testing. The data suggest that CRB2 present in rods is required to prevent photoreceptor degeneration and vision loss.

## 1. Introduction

The Crumbs protein complex is essential for polarity establishment and adhesion of the retinal neural epithelium. In mammals, the Crumbs family is composed of Crumbs homolog-1 (CRB1), CRB2, and CRB3. CRB1 and CRB2 have a large extracellular domain with epidermal growth-factor-like and laminin-A globular domains, a single transmembrane domain, and an intracellular C-terminal domain of 37 amino acids; CRB3 lacks the extracellular domain [1]. The intracellular domain has a single C-terminal PDZ protein-binding motif and a single FERM-protein-binding motif juxtaposing the transmembrane domain [1]. The Crumbs proteins associate with the adaptor protein PALS1 and one of the multiple PDZ-proteins PATJ or MUPP1 to form the core of the Crumbs complex [2,3]. In the developing mouse retina, the Crumbs proteins localize at the subapical region adjacent to the adherens junctions between retinal progenitor cells [4]. In the mature retina, the Crumbs proteins are present in photoreceptors and Müller glial cells (MGCs) [5,6]. CRB2 protein is present in photoreceptors and MGCs, whereas CRB1 protein is present only in MGCs [7,8].

Variations in the *CRB1* gene are associated with a wide variety of autosomal recessive retinal dystrophies, including retinitis pigmentosa (RP), Leber congenital amaurosis (LCA), cone-rod dystrophy, isolated macular dystrophy, and foveal retinoschisis [9,10]. So far, 310 pathogenic variations in the *CRB1* gene were identified (http://www.LOVD.nl/CRB1). Missense mutations in the human *CRB2* gene have been recently associated with RP [11], as well as with syndromic kidney and brain diseases [12,13,14].

Loss or reduced levels of the CRB1 or CRB2 proteins in retinal progenitors, immature photoreceptors, or MGCs leads to different retinal phenotypes in mice that mimic the wide spectrum of clinical features described in *CRB1*-patients, including early and late onset RP and LCA [2]. Loss of either CRB1 or CRB2 in MGCs results in mild retinal dystrophy, without impairing retinal function [6,15,16,17]. Ablation of *Crb2* in retinal progenitor cells, or in immature rod and cone photoreceptor cells, results in progressive thinning and degeneration of the photoreceptor layer, abnormal lamination of immature rods, and loss of retinal function [4,16,18]. Moreover, CRB2 has roles in restricting the proliferation of retinal progenitor cells and the number of rod photoreceptors and MGCs [4]. Mouse CRB2 acts as the modifying factor of *CRB1*-related retinal dystrophies, since reduction or full ablation of CRB2 in combination with loss of CRB1 results in an exacerbation of the retinal phenotype observed in *Crb1* knockout retinas [19,20,21,22]. The specific roles of CRB2 in rod photoreceptor cells still need to be elucidated. We hypothesize that CRB2 in rods is required to maintain proper retinal structure and function.

In fetal human retinas, human iPSC-derived retinal organoids, and adult non-human-primate retinas, CRB1 as well as CRB2 proteins localize at the subapical region in both photoreceptors and MGCs [22,23]. We previously showed proof-of-concept for AAV9-CMV-*CRB2*, reintroduction of CRB2 into photoreceptors and MGCs rescued the phenotype of *Crb2^flox/flox^*Chx10*Cre* and *Crb1Crb2^F/+^*Chx10*Cre* mouse retinas [24]. Although AAV9 efficiently transduces both mouse photoreceptors and MGCs [25], AAV5 outperforms AAV9 in transducing both human photoreceptors and MGCs in cultured adult human retinal explants and human iPSC-derived retinal organoids [23]; as such, AAV5 the most suitable serotype to be used in the clinics. In mice, AAV5 only infects retinal pigment epithelium and photoreceptors; a new animal model lacking CRB2 specifically in photoreceptor cells that allows us to test the efficacy of the AAV5-CMV-*CRB2* vector is required. Therefore, to validate our hypothesis and for the ability of testing the AAV5-CMV-*CRB2* vector in the future, we generated mice lacking CRB2 specifically in adult rod photoreceptors (with remaining levels of CRB2 in MGCs and cone photoreceptors), and mice lacking CRB2 specifically in rod photoreceptors and CRB1 specifically in MGCs.

Here, we studied the effects on retinal morphology and function of loss of CRB2 specifically in rod photoreceptors with or without concomitant loss of CRB1. Our data shows that specific ablation of CRB2 in mouse rods leads to RP. The phenotype observed in these retinas was characterized by loss of photoreceptor cells and gliosis in the peripheral and central retina. The retinal degeneration was more severe in the superior than in the inferior retina. Retinal function, measured by ERG, was impaired in 9-month-old animals. Concomitant loss of CRB1 exacerbates the retinal phenotype leading to decreased retinal and visual function from 3-months-of-age. The data suggest that CRB2 in rods is required to maintain cellular adhesion between rods and prevent photoreceptor degeneration and vision loss.

## 2. Results

### 2.1. Rho-iCre Mediates Recombination Specifically in Rod Photoreceptors

To study the specific cellular and physiological functions of CRB2 in rod photoreceptor cells, we crossed the *Crb2* floxed homozygous (*Crb2^flox^*^/*flox*^) mice [4,18] and *Crb1^KO^Crb2^flox^*^/flox^ [19] with the Rho-*iCre* transgenic mouse line [26] to obtain *Crb2^ΔRods^* (*Crb2^flox/flox^/*Rho*iCre*^+/−^) and *Crb1^KO^Crb2^ΔRods^* (*Crb1*^−/−^*Crb2^flox^*^/flox^/Rho*iCre*^+/−^) animals. The Rho-*iCre* transgenic mice express CRE recombinase in rod photoreceptors from postnatal day 7, resulting in efficient recombination at postnatal day 18 [26]. To confirm the mosaicism and specificity of the Rho-*iCre* transgenic mouse line, we crossed the *Crb2^ΔRods^* mice with a R26-stop-*EYFP* reporter mouse line [27] that expresses enhanced yellow fluorescent protein (EYFP) upon CRE-mediated recombination. While in *Crb2^flox^*^/flox^::*EYFP^flox^*^-stop-*flox*/+^ no EYFP signal could be detected in the retina (Appendix A), in *Crb2^ΔRods^*::*EYFP* mice, EYFP fluorescence was observed in the outer nuclear layer and the inner-segment layer of postnatal day 20 retinas (Appendix A). We checked for absence of CRE-mediated recombination in cone photoreceptors by performing co-localization studies using a cone photoreceptor marker (cone arrestin) and EYFP (Appendix A). Cone photoreceptors’ cell soma and inner-segments were EYFP-negative (Appendix A; arrows), suggesting that recombination mediated by Rho-*iCre* was efficient and limited to rod photoreceptors. CRB2 was detected at the subapical region at the outer limiting membrane in postnatal day 20 control (Appendix A) and *Crb2^ΔRods^* (Appendix A) retinas, likely due to maintained expression of CRB2 in the adjacent subapical region of wild-type MGCs and cone photoreceptors.

### 2.2. Ablation of CRB2 in Rods with Concomitant Loss of CRB1 Leads to Retinal Dysfunction and Vision Impairment

To study if the specific loss of CRB2 from rod photoreceptors with and without concomitant loss of CRB1 affected retinal function, we performed electroretinography (ERG) in 1-, 3-, 6-, 9-, and 12-month-old *Crb2^ΔRods^*, *Crb1^KO^Crb2^ΔRods^*, and the respective age-matched controls (*Crb2^flox^*^/*flox*^ and *Crb1^KO^Crb2^flox^*^/flox^). One-month-old *Crb2^ΔRods^* and *Crb1^KO^Crb2^ΔRods^* mice showed similar scotopic and photopic responses to the ones observed in the age-matched controls (Figure 1C, Appendix A). In contrast, 3-month-old *Crb1^KO^Crb2^ΔRods^*, but not *Crb2^ΔRods^* mice, showed slightly reduced amplitudes of a-wave scotopic electroretinography responses, indicating alterations of rod photoreceptor function (Figure 1A, arrow; Figure 1C). Moreover, 3-month-old *Crb1^KO^Crb2^ΔRods^* also showed reduced b-wave photopic electroretinography (Appendix A). The reduction of the scotopic a-wave amplitudes in *Crb1^KO^Crb2^ΔRods^* mice became more evident at 9- and 12-months-of-age (Figure 1B–D). At these time points, a reduced a-wave was also observed in *Crb2^ΔRods^* mice (Figure 1B–D; arrows). Twelve-month-old mutants also showed a reduced b-wave (Figure 1C).

Three-month-old *Crb1^KO^Crb2^ΔRods^* mice showed reduced scotopic a-wave responses, making these mice an interesting and potential suitable *CRB1*-disease model for future gene therapy rescue experiments. Therefore, we decided to determine whether the visual function was also affected in these mice. To do so, we used an optomotor response test (optokinetic tracking reflex (OKT)) to measure the spatial frequency threshold and contrast sensitivity [28,29]. Spatial frequency was measured by systematically increasing the spatial frequency of the grating at 100% contrast until animals no longer tracked (spatial frequency threshold). A contrast sensitivity threshold was generated by identifying the minimum contrast that generated tracking, over a range of spatial frequencies. Spatial frequency threshold and contrast sensitivity of *Crb1^KO^Crb2^ΔRods^* mice were analyzed at different time points, at 1-, 3-, 7-, and 9-month(s)-of-age. At 1- and 3-months-of-age, no differences were observed in spatial acuity between littermate controls (*Crb1^KO^Crb2^flox^*^/flox^) and *Crb1^KO^Crb2^ΔRods^* mice (Figure 2A). However, at 7- and 9-months-of-age the *Crb1^KO^Crb2^ΔRods^* showed a small but statistically significant decrease in spatial frequency thresholds, also described frequently in literature as visual acuity.

The retinas of one-month-old *Crb1^KO^Crb2^ΔRods^* showed similar contrast sensitivity compared to *Crb1^KO^Crb2^flox^*^/flox^ mice at all frequencies measured (Figure 2C, Appendix A). However, 3-, 7-, and 9-month-old *Crb1^KO^Crb2^ΔRods^* exhibited reduced contrast sensitivities compared to the littermate age-matched controls (*Crb1^KO^Crb2^flox^*^/flox^). Once *Crb1^KO^Crb2^ΔRods^* showed impaired visual function, we wanted to evaluate if deletion of only CRB2 from rods was sufficient to induce such a deficit. Therefore, the spatial frequency threshold and contrast sensitivity of *Cbr2^flox^*^/*flox*^ were compared to age-matched *Crb2^ΔRods^*. No differences were observed in spatial frequency threshold nor in contrast sensitivity (Figure 2B and Appendix A). The data suggest that the visual function impairment observed in the *Crb1^KO^Crb2^ΔRods^* is due to cumulative loss of CRB1 and CRB2 and not to single loss of CRB2 or due to toxicity of iCRE expression in rod photoreceptors.

### 2.3. Pupil Light Reflex is not Impaired in Crb1^KO^Crb2^ΔRods^ Mice

To assess if pupil light reflex was affected in *Crb1^KO^Crb2^ΔRods^* mice, we characterized the pupil response to blue and red light stimuli in dark-adapted, light-anesthetized 3- and 9-month-old *Crb1^KO^Crb2^ΔRods^* and in control mice. Pupil response curves, contraction, and dilation after each stimulus were identical in all the experimental groups (Figure 3A,C). No differences in the maximal pupil contraction were observed between mutant and control(s) mice at any condition analyzed (Figure 3B,D). These results suggest that pupil light reflex was not affected in the *Crb1^KO^Crb2^ΔRods^* mice.

### 2.4. Loss of CRB2 in Rods Results in Slow Loss of Photoreceptor Cells, Mainly in the Superior Retina

To study if CRB2 specific ablation in rod photoreceptors results in a morphological phenotype, histological analysis of retina sections was performed. All retinal layers were present and displayed normal organization in the *Crb2^ΔRods^* and *Crb1^KO^Crb2^ΔRods^* mice at 1-month-of-age, suggesting that removal of CRB2 from rod photoreceptors did not affect retinal development and lamination. No morphological abnormalities were observed in the control mice (*Crb2^flox/flox^*) (Figure 4A,A’) and in the *Crb2^ΔRods^* (Figure 4B,B’), while in the *Crb1^KO^Crb2^ΔRods^* (Figure 4C,C’; arrows), small disruptions of the outer limiting membrane were observed throughout the entire retina. Three-month-of-age retinas from control mice (*Crb2^flox/flox^*) (Figure 4D,D’) and in the *Crb2^ΔRods^* (Figure 4E,E’) showed normal morphology. Ingression of photoreceptor nuclei into the inner retinal layer was frequently detected in the inferior central retina of *Crb1^KO^Crb2^ΔRods^*, as also observed in the control littermate *Crb1^KO^Crb2^flox^*^/*flox*^ retinas [6,15,19] (data not shown). In *Crb1^KO^Crb2^ΔRods^*, disruptions at the outer limiting membrane of the central superior retina were also observed (Figure 4F,F’; arrow). In some 3-month-old and in all 6-month-old *Crb2^ΔRods^* (Figure 4H,H’; arrows) and *Crb1^KO^Crb2^ΔRods^* (Figure 4F,F’; arrows) retinas, the photoreceptor layer was severely thinned at the peripheral superior area. In 9- and 12-month-old *Crb2^ΔRods^* and *Crb1^KO^Crb2^ΔRods^*, further thinning of the outer nuclear layer was observed in the peripheral as well as central superior retina (Figure 5B,B’,C,C’,E,E’,E”,F,F’,F”).

To assess and better visualize the photoreceptor cell loss over time, the number of nuclei in rows of photoreceptors was quantified. The number of photoreceptor nuclei in a row was reduced in the entire superior retina and the inferior retina near to the optic nerve head in 6-month-old *Crb1^KO^Crb2^ΔRods^* compared to the *Crb2^flox^*^/*flox*^ control (Figure 5G). At 9-months-of-age, the number of photoreceptors in a row were further decreased in almost the entire superior retina and in most of the inferior retina of *Crb1^KO^Crb2^ΔRods^* (Figure 5H). Also at this time point, the superior retina (Figure 5H) and some areas of the inferior retina of *Crb2^ΔRods^* mice became thinner. Twelve-month-old retinas showed almost no photoreceptor nuclei in the far peripheral superior and nearly half in the central superior retina of *Crb2^ΔRods^* and *Crb1^KO^Crb2^ΔRods^*. At this time also, the inferior retina of *Crb2^ΔRods^* and *Crb1^KO^Crb2^ΔRods^* mice became thinner compared to the *Crb2^flox^*^/*flox*^ control (Figure 5I).

### 2.5. Removal of CRB2 in Rods Results in Loss of Rods Mainly at the Periphery of the Superior Retina

The morphological phenotype observed in the *Crb2^ΔRods^* and *Crb1^KO^Crb2^ΔRods^* retinas mainly affected the photoreceptor cells. In the far peripheral superior retinas of 9 month-old *Crb2^flox^*^/flox^ control mice, the photoreceptor cells stained positive for recoverin (Figure 6A). In the far periphery of the superior retinas of *Crb2^ΔRods^* (Figure 6B) and *Crb1^KO^Crb2^ΔRods^* (Figure 6C), no photoreceptors were found. Towards the central superior retina, reduced numbers of recoverin-positive cells were detected, depicting thinning of the outer nuclear layer. Although the number of photoreceptors was reduced, the remaining photoreceptors showed matured inner- and outer-segments. Rhodopsin is normally located in the outer-segments of mature rod photoreceptor cells (Figure 6D). In areas that showed reduced numbers of photoreceptors in *Crb2^ΔRods^* and *Crb1^KO^Crb2^ΔRods^* retina, rhodopsin was localized in the outer segments, suggesting that the remaining rods were functional (Figure 6E,F). Cone photoreceptors can be labelled using an antibody against cone arrestin (CAR) (Figure 6D). In both knockout lines, cone photoreceptors showed normal morphology (Figure 6E,F). In the control retinas, peanut-agglutinin (PNA) stained the cone photoreceptor outer-segments and pedicles at the photoreceptor synapses (Figure 6G). PNA staining was similar in 9-month-old *Crb2^ΔRods^* (Figure 6H) and *Crb1^KO^Crb2^ΔRods^* retinas (Figure 6I). The M-cone photoreceptor outer-segments were stained appropriately with M-opsin antibodies in the control retinas (Figure 6G) and in both mutant retinas (Figure 6H,I).

Photoreceptor cell synapses can be stained with MPP4. In the control retinas, MPP4 signal was detected in the photoreceptor synapses at the outer plexiform layer (Figure 6J,J’). In *Crb2^ΔRods^* and *Crb1^KO^Crb2^ΔRods^* retinas, MPP4 staining was disrupted and decreased; some ectopic anti-MPP4 labelling was also detected in the outer nuclear layer (Figure 6K,K’,L,L’). Protein kinase (PKC)α is abundant in retinal bipolar cells. In the control retinas, bipolar cells located at the inner nuclear layer and presented normal dendritic arborization in the outer plexiform layer (Figure 6J,J’). In the knockout retinas, bipolar cells were localized at the correct layer, but their dendritic arborization was affected (Figure 6K,K’,L,L’; arrowheads).

### 2.6. Loss of CRB2 in Rods Leads to Disruption of the Apical Protein Complexes

We previously reported that ablation of CRB2 from both immature cone and rod photoreceptor cells resulted in disruption of the apical protein complexes at the outer limiting membrane [16]. Here, we studied if removal of CRB2 specifically in rod photoreceptors is enough to lead to destabilization of the subapical region and adherens junction protein complexes at the outer limiting membrane using transmission electron microscopy and immunohistochemistry.

Using transmission electron microscopy, we showed that the structures of adherens junctions at the outer limiting membrane adjacent to photoreceptor cell nuclei were normal in retinas of 8 month-old control mice (Figure 7A, arrows; Figure 7A’), while the adherens junctions in the *Crb1^KO^Crb2^ΔRods^* mice were moderately disrupted throughout the retina (Figure 7B; arrows). Photoreceptor cell nuclei protruding the inner-/outer-segment layer were observed at sites of adherens junction disruption (Figure 7B’; blue line). In the central retina, there were more MGC apical villi visible in the *Crb1^KO^Crb2^ΔRods^* retina compared to the control retina (Figure 7C,D; asterisk). At the peripheral control retina, the MGC apical villi extend among the photoreceptor inner segments (Figure 7E; asterisk), whereas in the peripheral *Crb1^KO^Crb2^ΔRods^* retina, at regions with a diminished number of inner and outer segments, the MGC apical villi collapsed due to lack of support (Figure 7F; asterisk).

At 3-months-of-age, CRB1 protein was present at the subapical region above the adherens junctions in the control (Figure 8A; arrows) and *Crb2^ΔRods^* (Figure 8B; arrows) in the superior peripheral retina. In *Crb1^KO^Crb2^ΔRods^*, the CRB1 protein was absent, as previously found in *Crb1^KO^* (Figure 8C). CRB2 localized at the subapical region in the control retinas (Figure 8D) and in the central superior retina of *Crb2^ΔRods^* (Figure 8E) and *Crb1^KO^Crb2^ΔRods^* (Figure 8F). Other subapical region markers and members of the Crumbs complex, such as PALS1 and MUPP1, were correctly located at the subapical region in the control retina (Figure 8G). However, mutant retinas showed disruptions of the subapical region, labelled by CRB2, PALS1, and MUPP1 (Figure 8E,F,H,I; arrows). In the control retinas, β-catenin showed correct localization of the adherens junction (Figure 8J), while in both mutant retinas, disruptions of the adherens junctions in the central superior retina were observed (Figure 8K,L). Photoreceptor cell nuclei protruding from the inner-/outer-segment layer were observed at the site of disruption (Figure 8 arrows). In the 9-month-old control retina, CRB2 correctly localized at the subapical region just above the adherens junction marker, β-catenin (Figure 8M,M’). Also, PALS1 and MUPP1 were correctly located at the subapical region in the control retina (Figure 8P,P’). However, at the peripheral superior retina of *Crb2^ΔRods^* and *Crb1^KO^Crb2^ΔRods^* mice, partial disruptions of the subapical region were observed (Figure 8N,N’,O,O’,Q,Q’,R,R’), whereas adherens junctions were mainly lost in areas with loss of all photoreceptors (Figure 8N,N’,O,O’). PAR3, a member of the PAR complex, and p120-catenin, an adherens junction protein, were lost at sites of disruption of the outer limiting membrane in the central superior retina (Figure 8T,U; arrows).

### 2.7. Removal of CRB2 from Rods Results in Gliosis in Müller Glial Cells

Müller glial cells extend throughout the entire retina and function to maintain retinal homeostasis and integrity [30]. To study the effect of CRB2 removal from rod photoreceptor cells on the morphology of MGCs, we labelled these cells with glutamine synthetase (GS), SOX9, CD44, and glial fibrillary acidic protein expression (GFAP) antibodies. In 9 month-old control retinas, MGCs stained with glutamine synthetase displayed radial alignment, with well-established apical ends (Figure 9A,A’) and nuclei (SOX9-positive) located in the inner nuclear layer (Figure 9A,A’). In *Crb2^ΔRods^* and *Crb1^KO^Crb2^ΔRods^* retinas, the radial alignment of MGCs was disturbed and SOX9-positive nuclei were located most apically (Figure 9B,B’,C,C’; arrows). CD44 is highly expressed in MGC apical villi (Figure 9D,D’). In the peripheral superior knockout retinas, the radial structure of MGC apical villi was lost (Figure 9E,E’,F,F’). GFAP is a marker for intermediate filaments in MGCs (Figure 9G,J,M), and an increase in GFAP occurs in gliosis. Knockout retinas showed an upregulation of GFAP in MGCs (Figure 9H,I,K,L,N,O). Increased GFAP levels were more pronounced in the peripheral (Figure 9H,I) and central (Figure 9K,L) superior retina. A moderate increase in GFAP levels was also observed in the inferior retina of *Crb2^ΔRods^* and *Crb1^KO^Crb2^ΔRods^* (Figure 9N,O).

## 3. Discussion

Here, we studied the effect of CRB2 removal specifically from rod photoreceptors in the mouse retina. Furthermore, we evaluated the consequences of concomitant loss of CRB1. Our key findings are: (i) loss of CRB2 from rod photoreceptors results in retinitis pigmentosa, mainly at the peripheral and central superior retina; (ii) CRB2 in rod photoreceptor cells is required to maintain the rod photoreceptor layer and retinal electrical responses; (iii) loss of CRB2 in rods and concomitant loss of CRB1 leads to an exacerbation of the retinitis pigmentosa phenotype; (iv) ablation of CRB2 from rods with concomitant loss of CRB1 results in visual function impairment.

During retinogenesis, mouse CRB2 is located in retinal radial progenitor cells [4]. In the mature retina, the protein is found in rod and cone photoreceptors and in MGCs [7]. We previously demonstrated that retinal development and lamination is affected from embryonic day 15.5, when CRB2 is removed from both immature rod and cone photoreceptors. These *Crb2^ΔimmPRC^* mouse retinas mimicked a very early-onset retinitis pigmentosa (Appendix A) [16]. The *Crb2^ΔimmPRC^* mouse retinas showed progressive thinning of the photoreceptor layer and mislocalization of retinal cells, which resulted in a severe retinal function impairment, as measured by electroretinography. Moreover, concomitant loss of CRB1 in *Crb1^KO^Crb2^ΔimmPRC^* retinas exacerbated the retinal phenotype and resulted in an LCA phenotype with thickened superior retina due to abnormal lamination of photoreceptors, intermingled photoreceptor and inner nuclear cell nuclei, and ectopic photoreceptor nuclei in the ganglion cell layer [21].

The *Crb1^KO^Crb2^ΔRods^* retinas described here showed moderate decrease in retinal function and a significant decrease in contrast sensitivity shortly after the onset of morphological retinal degeneration at 3-months-of-age (Appendix A). In these mice, specific ablation of CRB2 from rod photoreceptors was achieved by crossing *Crb2^flox^*^/*flox*^ mice with Rho-*iCre* transgenic mice that express iCRE in rod photoreceptors from postnatal day 7, achieving expression in nearly all rods at postnatal day 20 [26]. The onset of CRE expression after retinogenesis might explain the moderate retinal phenotype observed in the *Crb2^ΔRods^* and *Crb1^KO^Crb2^Δrods^* mice when compared to mice with Cre-mediated ablation in retinal progenitor cells (*Crb2^ΔRPC^*) or mice with Cre-mediated ablation in immature cone and rod photoreceptor cells (*Crb2^ΔimmPRC^*). We previously demonstrated that short-term depletion of CRB2 from adult retinas using adeno-associated viral delivery of *Cre* or shRNA against *Crb2* leads to sporadic disruptions at foci of the outer limiting membrane, outer plexiform layer, and disruption of adhesion between photoreceptor cells [16]. This, together with the current data reported here, suggests that CRB2 has a function in the regulation of cellular adhesion between rod photoreceptors. Several *CRB1*-disease mouse models showed a gradient in phenotype severity. The entire retina is affected in *Crb2^ΔRPC^* mice lacking CRB2 from retinal progenitors, MGCs, and photoreceptor cells, or *Crb2^ΔimmPRC^* mice lacking CRB2 from the immature rod and cone photoreceptors [4,16]. In *Crb1^KO^* retinas lacking CRB1 in retinal progenitors and MGCs, the phenotype is mainly located in the inferior temporal quadrant [6,15]. *Crb1^KO^Crb2^ΔimmPRC^* retinas lacking CRB1 and CRB2 from immature photoreceptors showed retinal dystrophy by fusion of the inner and outer nuclear layer throughout the retina. However, larger regions of ectopic photoreceptor cells were observed in the superior retina [21]. The *Crb2^ΔRods^* and *Crb1^KO^Crb2^ΔRods^* mice reported here presented a retinal phenotype mainly affecting the peripheral and central superior retina. The differences observed in the different mouse *CRB1*-models might be related to the higher levels of CRB2 at the subapical region in the inferior retina, whereas CRB1 is expressed at higher levels in the superior retina [20]. Several other genes were found to be enriched in the superior retina, amongst these the photoreceptor gene endothelin 2 (Edn2) and the MGC genes ceruloplasmin and glial fibrillary acidic protein (Gfap) [31]. In the *Crb2^ΔRods^* and *Crb1^KO^Crb2^ΔRods^* mice, GFAP expression was upregulated in the superior retina, with structural changes in the subapical villi of MGC, which suggests the contribution of MGCs to the phenotype observed. Retinal function mediates visual perception, but also other vision-linked reflexes, such as the pupillary light reflex [32,33,34]. Previously, Kostic et al. demonstrated that low-intensity red or blue light stimulus was a good parameter to discriminate rod photoreceptor contribution for the pupil response, while the recovery time after high-intensity blue stimulus was a predictor cone contribution to the pupillary light reflex [33]. Here, we used a vision-linked reflex, the pupillary light reflex, as a retina/vision functional outcome. *Crb1^KO^Crb2^ΔRods^* mice showed a strong reduction in contrast sensitivity from 3-months-of-age onwards, which was not observed in *Crb2^ΔRods^* mice, and a small decrease in spatial frequency from 7-months-of-age onwards, while the control *Crb1^KO^Crb2^flox^*^/*flox*^ mice did not show a reduction at 9-months-of-age. To our knowledge, this is the first report of loss of visual function as measured by OKT in a *CRB1*-retinal disease model. The data suggest that visual contrast sensitivity might be a diagnostically accurate and practical test to detect early visual deficit in *Crb1^KO^Crb2^ΔRods^* mice. Moreover, contrast sensitivity tests might potentially be suitable for clinical studies testing candidate medicines for patients with mutations in the *CRB1* gene. The correlation between photoreceptor degeneration and OKT contrast sensitivity in different mouse models are poorly described in the literature. However, some studies used different rhodopsin mutant rat strains and demonstrated that OKT spatial frequency and contrast sensitivity did not decline until very late in the photoreceptor degeneration [35,36]. Furthermore, studies performed in mice carrying a missense point mutation in *Pde6b* (*rd10*), which are almost deprived of photoreceptor cells at 1-month-of-age, still presented residual spatial frequency responses at 2-months-of-age [37].

Here, we propose that loss of CRB2 leads to loss of rod-cone and rod-MGC adhesion; as a result adherens junctions are disrupted and the cytoarchitecture of the outer nuclear layer is compromised, giving rise to misplaced rod photoreceptor cells. We speculate that misplaced or “loose” rods preferentially degenerate and die or are phagocyted by activated microglial cells. Microglia cell activation will contribute to MGC reactive gliosis, characterized by increased GFAP levels and hypertrophy, by secreting pro-inflammatory mediators. We further speculate that the reactive MGCs from *Crb1^KO^Crb2^ΔRods^* and *Crb2^ΔRods^* mice present a proliferation of fibrous processes and deposition of proteoglycans at the outer edge of the retina, inhibiting axonal regeneration and exacerbating the loss of rods (Appendix A) [38]. Moreover, the clear deficit of contrast sensitivity may allow us to have a rigorous outcome parameter to measure functional vision gain or maintenance after AAV treatment. Therefore, *Crb1^KO^Crb2^ΔRods^* mice might become important resources to test therapies for retinopathies due to mutations in the *CRB1* gene.

## 4. Materials and Methods

### 4.1. Animals

All procedures concerning animals were carried out in accordance with the E.U. Directive 2010/63/EU for animal experiments and with permission from the Dutch Central Authority for Scientific Procedures on Animals (CCD), permit number 1160020172924. All mice used were maintained with a 99.9% C57BL/6J genetic background. Animals were kept in a 12 hours day/night cycle and supplied with food and water ad libitum. Mice did not have *rd8* or *pde6b* mutations.

*Crb2* conditional knockout (*Crb2^flox/flox^*) mice [4,18] were crossed with the Rho-iCre (B6;SJL-Pde6b+Tg(Rho-iCre)1Ck/Boc) [26] transgenic mouse line, obtained from Jackson laboratory (IMSR Cat# JAX:015850, RRID: IMSR_JAX:015850), to specifically remove *Crb2* from rod photoreceptor cells (*Crb2^flox/flox^/Rho-iCre^+/^*^−^, mentioned as *Crb2^ΔRods^*). C*rb2^ΔRods^* were crossed with *Crb1* knockout (*Crb1^KO^*) mice [6] to generate *Crb1*^−*/*−^*Crb2^flox/flox^/Rho-iCre^+/^*^−^ (*Crb1^KO^Crb2^ΔRods^*). Littermate age-matched *Crb2^flox/flox^* or *Crb1^KO^Crb2^flox/flox^* were used as control. R26R-EYFP (B6.129X1-Gt(ROSA)26Sor^tm1(EYFP)Cos^/J) (IMSR Cat# JAX:006148, RRID:IMSR_JAX:006148) reporter mice [27] were crossed with *Crb2^ΔRods^* to generate *Crb2^ΔRods^ EYFP^flox-stop-flox/+^* (*Crb2^ΔRods^::EYFP*) mice.

### 4.2. Chromosomal DNA Isolation and Genotyping

Chromosomal DNA was isolated from ear biopsies. The biopsies were incubated in lysis buffer (50 mM Tris pH 8.0, 100 mM NaCl, 1% SDS) with Proteinase K (0.5 mg/ml) at 55 °C for 16 h. Chromosomal DNA was precipitated with isopropanol, washed with 80% ethanol, and rehydrated in TE buffer. Genotyping of the *Crb1^KO^* and *Crb2^flox^*^/*flox*^ transgenic animals were performed as previously described [4,6]. The following primers were used to detect the transgenic *iCre* expression: FW 5′-TCAGTGCCTGGAGTTGCGCTGTGG-3′ and RV 5′-CTTAAAGGCCAGGGCCTGCTTGGC-3′ (product size 650 base-pairs). R26R-EYP mice were genotyped used the following primers: wild-type allele FW 5′-CTGGCTTCTGAGGACCG-3′, RV 5′-CAGGACAACGCCCACACA-3′ (product size 142 base-pairs), and mutant allele FW 5′-AGGGCGAGGAGCTGTTCA-3′, RV 5′-TGAAGTCGATGCCCTTCAG-3′ (product size 384 base-pairs).

### 4.3. Electroretinography (ERG)

Dark- and light-adapted ERGs were performed under dim red light using an Espion E2 (Diagnosys, LLC, Lowell MA, USA). ERGs were performed on 1-month-old (1M), 3M, 6M, 9M, and 12M in *Crb2^ΔRods^*, *Crb1^KO^Crb2^ΔRods^*, *Crb2^flox^*^/*flox*^, and *Crb1^KO^Crb2^flox^*^/*flox*^ mice. Mice were anesthetized using 100 mg/kg ketamine and 10 mg/kg xylazine administrated intraperitoneally, and the pupils were dilated using tropicamide drops (5 mg/mL). Mice were placed on a temperature regulated heating pad and reference and ground platinum electrodes were placed subcutaneously in the scalp and the base of the tail, respectively. ERGs were recorded from both eyes using gold wire electrodes. Hypromellose eye drops (3 mg/mL, Teva, The Netherlands) were given between recordings to prevent eyes from drying. Single (Scotopic and Photopic ERG) white (6500 k)-flashes were used. Band-pass filter frequencies were 0.3 and 300 Hz. Scotopic recordings were obtained from dark-adapted animals at the following light intensities: −4, −3, −2, −1, 0, 1, 1.5, 1.9 log cd·s/m^2^. Photopic recordings were performed following 10 minutes light adaptation on a background light intensity of 30 cd·m^2^ and the light intensity series used was: −2, −1, 0, 1, 1.5, 1.9 log cd·s/m^2^ [39]. The following numbers of mice were used per time point: 1-month-old (1M): (*Crb2^flox^*^/*flox*^ = 4, *Crb1^KO^Crb2^flox^*^/*flox*^ = 5, *Crb2^ΔRods^* = 5, and *Crb1^KO^Crb2^ΔRods^* = 4); 3M: (*Crb2^flox^*^/*flox*^ = 5, *Crb1^KO^Crb2^flox^*^/*flox*^ = 6, *Crb2^ΔRods^* = 5, and *Crb1^KO^Crb2^ΔRods^* = 5); 6M: (*Crb2^flox^*^/*flox*^ = 6, *Crb1^KO^Crb2^flox^*^/*flox*^ = 5, *Crb2^ΔRods^* = 7, and *Crb1^KO^Crb2^ΔRods^* = 7); 9M: (*Crb2^flox^*^/*flox*^ = 5, *Crb1^KO^Crb2^flox^*^/*flox*^ = 4, *Crb2^Δrods^* = 5, and *Crb1^KO^Crb2^ΔRods^* = 6); 12M: (*Crb2^flox^*^/*flox*^ = 6, *Crb1^KO^Crb2^flox^*^/*flox*^ = 6, *Crb2^ΔRods^* = 6, and *Crb1^KO^Crb2^ΔRods^* = 4). Statistical analysis of the ERG data was performed using a t-test [22]. Responses of the *Crb1^KO^Crb2^ΔRods^* and *Crb2^ΔRods^* were compared to the *Crb2^flox^*^/*flox*^ control mice at each time point analyzed.

### 4.4. Pupillary Light Reflex

The mice were dark-adapted for 16 hours prior to experiments. Pupillary light reflex was measured under dim red light using an A2000 pupillometer (Neuroptics, Inc., Irvine, CA, USA). Pupillary light reflex was performed on 3-month-old (3M) *Crb1^KO^Crb2^flox^*^/*flox*^ and *Crb1^KO^Crb2^ΔRods^* mice and on 9-month-old *Crb2^flox^*^/*flox*^, *Crb1^KO^Crb2^flox/flox^*, and *Crb1^KO^Crb2^ΔRods^* mice. Mice were anesthetized with 50 mg/kg ketamine and 5 mg/kg xylazine injected intraperitoneally. Mice were placed on the platform, and pupil’s borders were placed in focus. The following light sequence was used: red 1.2 log·lux (−1.2 log·W/m^2^, 0.065 W/m^2^), blue 0.6 log·lux (−1.1 log·W/m^2^, 0.074 W/m^2^), red 4.5 log·lux (2.1 log·W/m^2^, 129.018 W/m^2^), blue 2.0 log·lux (0.3 log W/m^2^, 1.893 W/m^2^) [33]. Pupil diameter recording started 500 ms before the light stimulus and continued for 29 seconds after the light stimulus. The pupil diameter was determined automatically by software from Neuroptics, Inc.

Pupil light reflex measurement amounted to 320 stimuli measurements (4 stimuli per pupil), of which 17 were removed after manual curation. The numbers of animals used were as follows: 3M *Crb1^KO^Crb2^flox^*^/*flox*^ (*n* = 56 stimuli, *n* = 8 mice), *Crb1^KO^Crb2^ΔRods^* (*n* = 67 stimuli, *n* = 9 mice) and 9M *Crb2^flox^*^/*flox*^ (*n* = 44 stimuli, *n* = 6 mice), *Crb1^KO^Crb2^flox^*^/*flox*^ (*n* = 64 stimuli, *n* = 8 mice), and *Crb1^KO^Crb2^ΔRods^* (*n* = 72 stimuli, *n* = 9 mice). All stimuli were normalized in percentile by the maximum dilation during the first 500 ms prior to stimulus and smoothed by running median (k = 5). The aggregated mean per group is shown in Figure 4A,C, and the overall maximum pupil constriction over 303 stimuli are shown in Figure 4B,D.

### 4.5. Optokinetic Tracking Reflex (OKT)

Spatial frequency and contrast sensitivity thresholds were measured using an OptoMotry system (Cerebral Mechanics, Lethbridge, AB, Canada). One-, 3-, 7-, and 9-month-old *Crb1^KO^Crb2^ΔRods^* and *Crb1^KO^Crb2^flox^*^/*flox*^ mice, and 9-month-old *Crb2^flox^*^/*flox*^ mice were placed on a small platform in the center of four computer monitors that formed a virtual drum with a rotating vertical sine wave grating (12°/s (d/s)), as described previously [29]. Reflexive head movements in the same direction as the rotating gratings were considered positive responses. Spatial frequency thresholds were determined with an increasing staircase paradigm starting at 0.042 cycles/deg (c/d) with 100% contrast. Contrast sensitivity thresholds were measured across six spatial frequencies (0.031, 0.064, 0.092, 0.103, 0.192, and 0.272 c/d). The reciprocal of the contrast sensitivity threshold was used as the contrast sensitivity value at each spatial frequency.

The numbers of mice used for the OKT measurements per time point were as follows: 1M (*Crb1^KO^Crb2^flox^*^/*flox*^ = 8 and *Crb1^KO^Crb2^ΔRods^* = 4), 3M (*Crb1^KO^Crb2^flox^*^/*flox*^ = 10 and *Crb1^KO^Crb2^ΔRods^* = 10), 5M (*Crb2^flox^*^/*flox*^ = 10, *Crb2^ΔRods^* = 8), 7M (*Crb1^KO^Crb2^flox^*^/*flox*^ = 8 and *Crb1^KO^Crb2^ΔRods^* = 9), 9M (*Crb2^flox^*^/*flox*^ = 7, *Crb1^KO^Crb2^flox^*^/*flox*^ = 8, and *Crb1^KO^Crb2^ΔRods^* = 9). Statistical significance was calculated by using Mann-Whitney U test.

### 4.6. Morphological and Immunohistochemical Analysis

Eyes from *Crb2^Δrods^, Crb1^KO^Crb2^Δrods^, Crb1^KO^Crb2^flox/flox^*, and *Crb2^flox/flox^* mice were collected at different time points: 1-month-old (1M), 3M, 6M, 9M, and 12M (*n* = 4–6/age/group). For morphological analysis, the superior part of the eye was marked with a yellow dye (Davidson Marking System® dyes, Bradley Products, Bloomington, MN, USA). Thereafter, eyes were enucleated and fixed at room temperature with 4% paraformaldehyde in phosphate buffered saline (PBS) for 20 min. After fixation, the eyes were dehydrated for 30 min in 30%, 50%, 70%, 90%, and 96% ethanol, embedded in Technovit 7100 (Kulzer, Wehrheim, Germany), and sectioned (3 µm), as previously described [40]. Slides were dried, counterstained with 0.5% toluidine blue, and mounted under coverslips using Entellan (Merk, Darmstadt, Germany). Eye sections were scanned using a Pannoramic 250 digital slide scanner (3DHISTECH Ltd., Budapest, Hungary) and images were processed with CaseViewer 2.1 (3DHISTECH Ltd., Budapest, Hungary).

For immunohistochemical analysis, eyes were enucleated and fixed for 20 min in 4% paraformaldehyde in PBS. Subsequently, the tissues were cryo-protected with 15% and 30% sucrose in PBS, embedded in Tissue-Tek O.C.T Compound (Sakura,Finetek, Alphen aan den Rijn, The Netherlands), and used for cryosectioning. Cryosections (8 µm) were rehydrated in PBS and blocked for 1 h using 10% goat serum, 0.4% Triton X-100, and 1% bovine serum albumin (BSA) in PBS. The primary antibodies were diluted in 0.3% goat serum, 0.4% Triton X-100, and 1% BSA in PBS, and incubated for 16 hours at 4 °C. Fluorescent-labelled secondary antibodies were goat anti-mouse, goat anti-rabbit, goat anti-chicken, or goat anti-rat IgGs conjugated to Alexa 488, Alexa 555 (1:1000; Abcam, Cambridge, UK), or Cy3 (1:500) were diluted in 0.1% goat serum in PBS and incubated for 1 hour at room temperature. Nuclei were counterstained with DAPI and mounted in Vectashield Hardset mounting medium (H1500, Vector Laboratories, Burlingame, CA, USA). Sections were imaged on a Leica TCS SP8 confocal microscope. Confocal images were processed with Leica Application Suite X (v3.3.0.16799) or Adobe Photoshop CC 2018 (Adobe Photoshop, RRID: SCR_014199).

### 4.7. Transmission Electron Microscopy

Eyes from 8-month-old *Crb1^KO^Crb2^ΔRods^* and respective control (*Crb2^flox^*^/*flox*^) were fixed in 4% paraformaldehyde and 2% glutaraldehyde in phosphate buffer (PB) for 24 h (*n* = 2 per group). Eyes were cut in half along the mid-sagittal plane using a razor knife, cutting through the optic nerve. The vitreous body was removed, and the two halves were placed back in the fixative for 2 hours. After rinsing, eyes were post-fixed in 1% OsO_4_/0.1M cacodylate buffer, rinsed again, and dehydrated with an ascending series of ethanol, followed by mixtures of propylene oxide and EPON (LX112). After the infiltration step with pure EPON, the halves of the eyes were positioned on the caps of large BEEM capsules filled with EPON. Ultrathin sections of the eyes, 80 nm thick, were made and stained with uranyl acetate and lead citrate and examined with a FEI Tecnai electron microscope (FEI Tecnai T12 Twin Fei Company, Eindhoven, The Netherlands; camera by OneView, Gatan) operating at 120 Kv. Overlapping images were collected and stitched together into separate images, as previously described [41].

### 4.8. Primary Antibodies

The following primary antibodies were used: β-catenin (1:250; BD Biosciences Cat# 610153, RRID:AB_397554), catenin p120 (1:250; BD Biosciences Cat# 610134 RRID:AB_397537), CD44 (1:400; BD Biosciences Cat# 553132, RRID:AB_394647), cone arrestin (1:500; Millipore Cat# AB15282, RRID:AB_1163387), CRB1 AK2 (1:200; homemade) [6], CRB2 SK11 (1:200; homemade), glial fibrillary acidic protein (GFAP) (1:200; Dako Cat# Z0334, RRID:AB_10013382), GFP (Millipore Cat# MAB3580, RRID:AB_94936), glutamine synthetase (GS) (1:250; BD Biosciences Cat# 610518, RRID:AB_397880), MPP4 AK4 (1:300; homemade) [6], MPP5/PALS1 SN47 (1:200; homemade), MUPP1 (1:200; BD Biosciences Cat# 611558, RRID: AB_399004), M-Opsin (1:250; Millipore Cat# AB5405, RRID: AB_177456). PAR3 (1:100; Millipore Cat# 07-330, RRID: AB_2101325), PKCα (1:250; BD Biosciences Cat# 610107 RRID: AB_397513), peanut agglutinin (PNA) (1:200; Vector Laboratories Cat# RL-1072, RRID: AB_2336642), recoverin (1:500; Millipore Cat# AB5585, RRID: AB_2253622), rhodopsin (1:500; Millipore Cat# MAB5356, RRID: AB_2178961), SOX9 (1:250; Millipore Cat# AB5535, RRID: AB_2239761).

### 4.9. Quantification of the Number of Photoreceptor Nuclei in a Row

The number of photoreceptor cell nuclei in a row were measured every 250 μm from the optic nerve head (ONH) in 1-month-old (1M), 3M, 6M, 9M, and 12M *Crb2^ΔRods^*, *Crb1^KO^Crb2^ΔRods^*, *Crb1^KO^Crb2^flox^*^/*flox*^ and *Crb2^flox^*^/*flox*^ mice. Three sections in the optic nerve area of 3 retinas from 3–4 independent mice per group were used for the measurements. Retina sections were scanned using a Pannoramic 250 digital slide scanner (3DHISTECH Ltd., Budapest, Hungary) and measurements were performed with CaseViewer 2.1 (3DHISTECH Ltd., Budapest, Hungary). The values of different sections of individual mice were averaged. The number of photoreceptor nuclei from the *Crb2^ΔRods^* and *Crb1^KO^Crb2^ΔRods^* were compared to the *Crb2^flox^*^/*flox*^ control mice.

### 4.10. Statistical Analysis

Normality of the distribution was tested by the Kolmogorov–Smirnov test. Statistical significance was calculated by using an unpaired t-test or by using Mann-Whitney U test if the data did not show a normal distribution. All statistical analyses were performed using GraphPad Prism version 7.02 (GraphPad Prism, RRID: SCR_002798). All values are expressed as mean ± SEM. Statistically significant values: * *p* < 0.05, ** *p* < 0.01, *** *p* < 0.001, **** *p* < 0.0001.

## 5. Conclusions

In conclusion, we found that (i) loss of CRB2 from rod photoreceptors results in retinitis pigmentosa, mainly at the peripheral and central superior retina; (ii) CRB2 in rod photoreceptor cells is required to maintain the rod photoreceptor layer and retinal electrical responses; (iii) loss of CRB2 in rods and concomitant loss of CRB1 leads to an exacerbation of the retinitis pigmentosa phenotype; (iv) ablation of CRB2 from rods with concomitant loss of CRB1 results in visual function impairment. This clear deficit of contrast sensitivity may allow us to have a rigorous outcome parameter to measure functional vision gain or maintenance after AAV treatment. Therefore, *Crb1^KO^Crb2^ΔRods^* mice might become important resources to test therapies for retinopathies due to mutations in the *CRB1* gene.

## 6. Patents

The Leiden University Medical Center (LUMC) is the holder of patent application PCT/NL2014/050549, which describes the potential clinical use of CRB2. J.W. is listed as inventor on this patent, and J.W. is an employee of the LUMC.

## Figures and Tables

**Figure 1 ijms-20-04069-f001:**
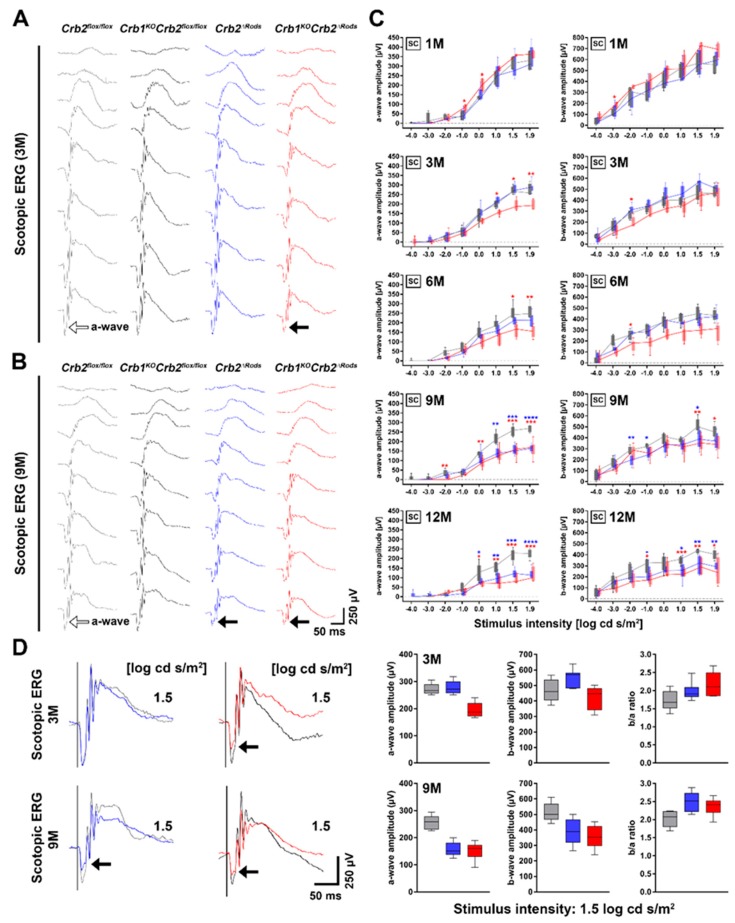
Progressive loss of retinal function in *Crb2^ΔRods^* and *Crb1^KO^Crb2^ΔRods^* mice. Electroretinographic analysis of retinal function in *Crb2^flox^*^/flox^ (control, gray), *Crb1^KO^Crb2^flox^*^/flox^ (control, black), *Crb2^ΔRods^* (blue), and *Crb1^KO^Crb2^ΔRods^* (red). Scotopic single-flash intensity series from representative animals at 3-months-of-age (**A**) and 9-months-of-age (**B**). The control scotopic a-wave is indicated by the open arrow and the black arrow points to the attenuated a-wave of the *Crb1^KO^Crb2^ΔRods^* at 3-months-of-age and of the *Crb1^KO^Crb2^ΔRods^* and *Crb2^ΔRods^* at 9-months-of-age (**A**,**B**; arrows). Time course single-flash ERG data from 1-, 3-, 6-, 9-, and 12-month-old *Crb2^flox^*^/flox^ (control), *Crb2^ΔRods^* and *Crb1^KO^Crb2^ΔRods^* mice (**C**). Scotopic (SC) a-wave and b-wave amplitude are presented as a function of the logarithm of the flash intensity. Superposition of scotopic single-flash electroretinography responses (1.5 log cd·s/m^2^) where the black arrow points to the attenuated a-wave of the affected mice (**D**) (left), and the quantitative evaluation as well as the corresponding b-wave/a-wave amplitude ratio (b/a ratio) (**D**) (right). Boxes indicate the 25% and 75% quantile range and whiskers indicate the 5% and 95% quantiles, and the intersection of line and error bars indicates the median of the data (box-and-whisker plot). * *p* < 0.05, ** *p* < 0.01, *** *p* < 0.001, **** *p* < 0.0001.

**Figure 2 ijms-20-04069-f002:**
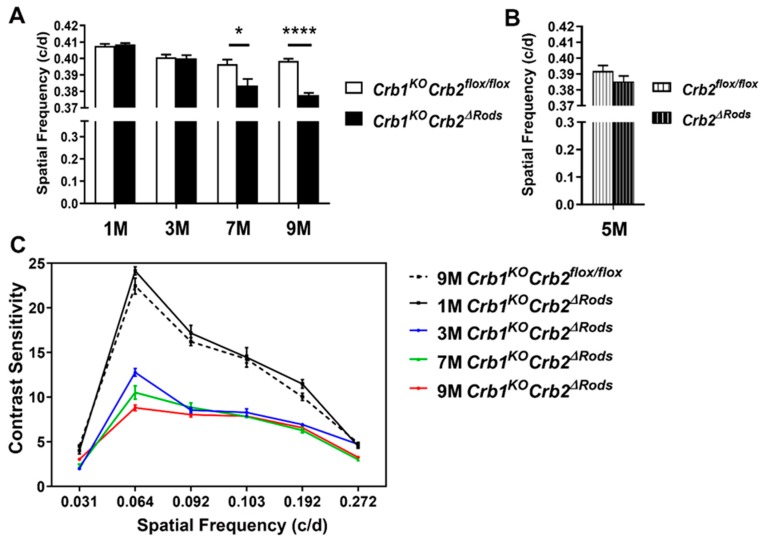
Visual acuity and contrast sensitivity are impaired in *Crb1^KO^Crb2^ΔRods^* mice. Spatial frequency thresholds (**A**,**B**) in cycles per degree (c/d) and contrast sensitivity (**C**) at different time points (1-month of age (1M), 3M, 5M, 7M, and 9M). Spatial frequency thresholds were reduced in 7- and 9-month-old *Crb1^KO^Crb2^ΔRods^* mice compared to the littermates and age-matched controls (*Crb1^KO^Crb2^flox^*^/flox^), at *p* = 0.0274 and *p* < 0.001, respectively (**A**). No differences were observed between *Crb2^flox^*^/flox^ and *Crb2^ΔRods^* mice at 5M (**B**). *Crb1^KO^Crb2^ΔRods^* mice showed reduced contrast sensitivity compared to the littermates, and age matched controls (*Crb1^KO^Crb2^flox^*^/flox^) at 3-, 7-, and 9-months-of-age (**C**). Contrast sensitivity at: 3M (spatial frequencies: 0.031, *p* = 0.7959 (statistically non-significant); 0.064, *p* < 0.0001; 0.092, *p* < 0.0001; 0.103, *p* < 0.0001; 0.192, *p* < 0.0001; 0.272, *p* = 0.0006); 7M (spatial frequencies: 0.031, *p* = 0.0121; 0.064, *p* = 0.0003; 0.092, *p* < 0.0001; 0.103, *p* = 0.004; 0.192, *p* = 0.0006; 0.272, *p* = 0.0448); and 9M (spatial frequencies: 0.031, *p* < 0.0001 (statistically non-significant); 0.064, *p* < 0.0001; 0.092, *p* = 0.0002; 0.103, *p* < 0.0001; 0.192, *p* < 0.0001; 0.272, *p* < 0.0001). Error bars indicate ± SEM, * *p* < 0.05, **** *p* < 0.0001.

**Figure 3 ijms-20-04069-f003:**
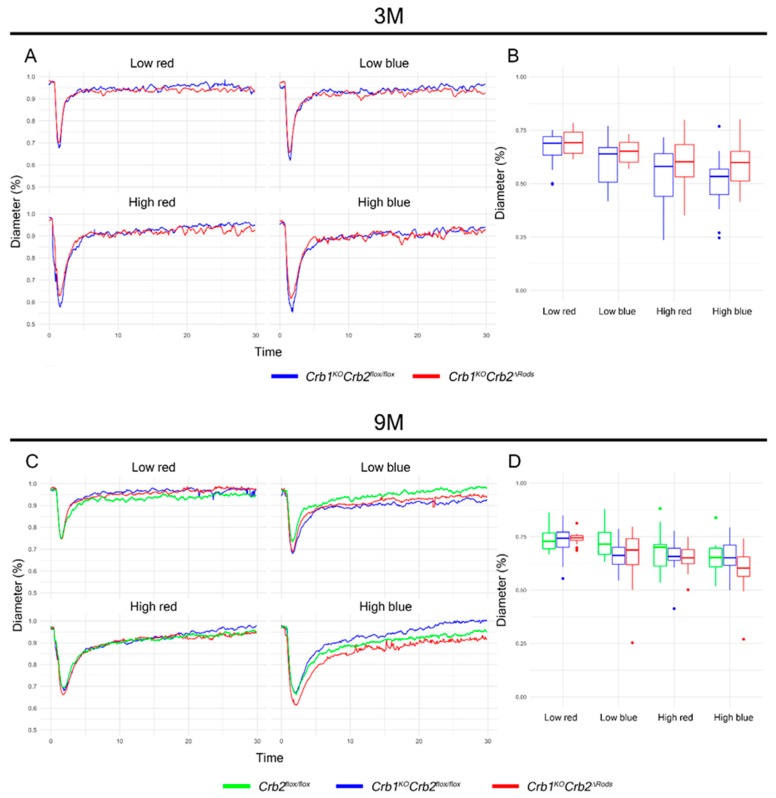
Pupil light reflex is not impaired in *Crb1^KO^Crb2^ΔRods^* mice. Pupil responses of 3-month-old *Crb1^KO^Crb2^flox^*^/flox^ (blue line) and *Crb1^KO^Crb2^ΔRods^* (red line) (**A**,**B**) and 9-month-old *Crb2^flox^*^/flox^ (green line), *Crb1^KO^Crb2^flox^*^/flox^ (blue line), and *Crb1^KO^Crb2^ΔRods^* (red line) mice (**C**,**D**) to chromatic light stimulation (500 ms red or blue light) of varying intensity: low red, 1.2 log·lux; low blue, 0.6 log·lux; high red, 4.5 log·lux; high blue, 2.0 log·lux. X-axis: time in seconds; Y-axis: average of pupil diameter (in %) relative to baseline diameter determined from 500 ms before each stimulus (**A**,**C**). Maximal pupil contraction (**B**,**D**), boxes indicate the 25 and 75% quantile range and whiskers indicate the 5% and 95% quantiles, and the intersection of line and error bars indicates the median of the data (box-and-whisker plot).

**Figure 4 ijms-20-04069-f004:**
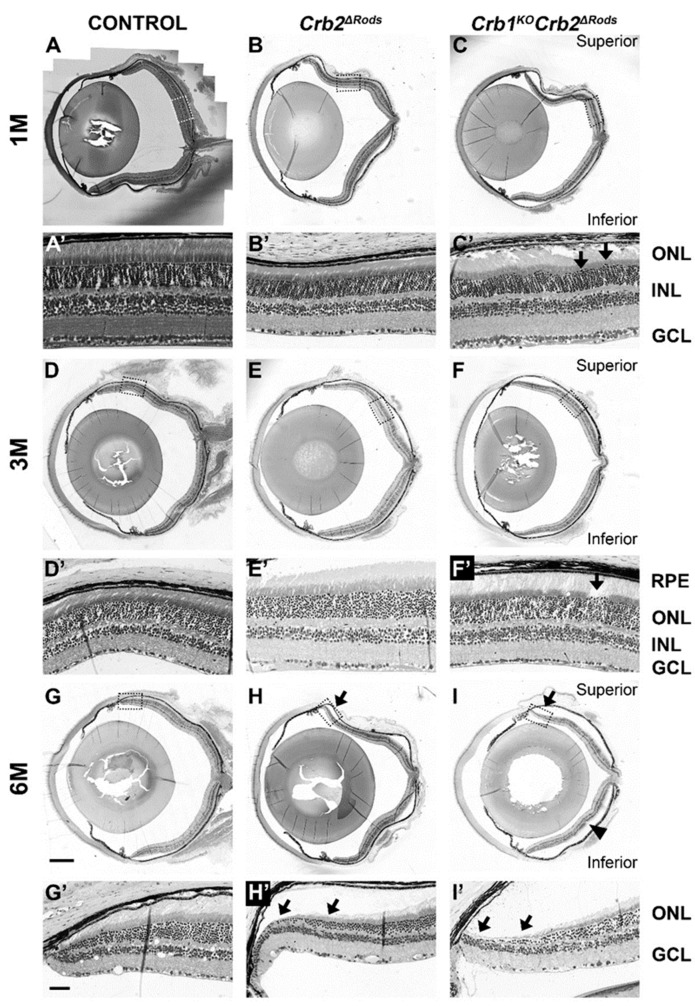
Loss of *Crb2* from rod photoreceptors leads to loss of photoreceptor cells, mainly in the peripheral and central superior retina. Toluidine-stained light microscopy showing retinal stitches from control (*Crb2^flox^*^/*flox*^) (**A**,**D**,**G**), *Crb2^ΔRods^* (**B**,**E**,**H**) and *Crb1^KO^Crb2^ΔRods^* (**C**,**F**,**I**), and insets from control (*Crb2^flox^*^/*flox*^) (**A’**,**D’**,**G’**), *Crb2^ΔRods^* (**B’**,**E’**,**H’**), and *Crb1^KO^Crb2^ΔRods^* (**C’**,**F’**,**I’**) at different ages: (**A**–**C**) 1M; (**D**–**F**) 3M; (**G**–**I**) 6M. No abnormalities were observed in the control retina. In the *Crb2^ΔRods^* and *Crb1^KO^Crb2^ΔRods^* mice, all the retinal layers were formed and properly laminated. At 3-months-of-age, sporadic disruptions of the outer limiting membrane were found in the *Crb1^KO^Crb2^ΔRods^* in the superior retina (**F,F’**; arrow). The characteristic *Crb1^KO^* phenotype, with photoreceptor dysplasia specifically in the inferior temporal retina, was also frequently observed in the inferior retina (**I**; arrowhead). At 6M, loss of photoreceptor cells was observed at the superior peripheral retina in both *Crb2^ΔRods^* (**H,H’**; arrows) and *Crb1^KO^Crb2^ΔRods^* (**I,I’**; arrows) mice. Note: GCL = ganglion cell layer; INL = inner nuclear layer; ONL = outer nuclear layer; RPE = retina pigment epithelium. Scale bars: 500 µm; insets: 50 µm.

**Figure 5 ijms-20-04069-f005:**
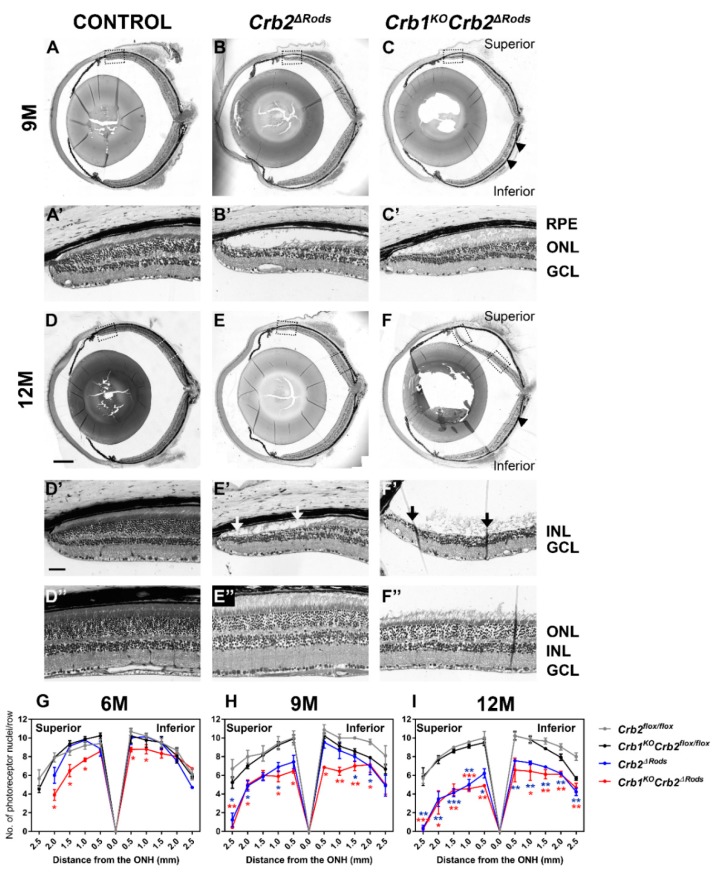
Aged *Crb2^ΔRods^* and *Crb1^KO^Crb2^ΔRods^* show loss of photoreceptors in the entire retina. Toluidine-stained light microscopy showing retinal stitches from control (*Crb2^flox^*^/*flox*^) (**A**,**D**), *Crb2^ΔRods^* (**B**,**E**), and *Crb1^KO^Crb2^ΔRods^* (**C**,**F**), and insets from control (*Crb2^flox^*^/*flox*^) (**A’**,**D’**,**D’’**), *Crb2^ΔRods^* (**B’**,**E’**,**E’’**), and *Crb1^KO^Crb2^ΔRods^* (**C’**, **F’**,**F’’**) at different ages: (**A**–**C**) 9M; (**D**–**F**) 12M. The peripheral superior retina is depicted (**D’,E’,F’**); the central superior retina is depicted (**D”,E”,F”**). No abnormalities were observed in the control retina. At 9- and 12-months-of-age, loss of photoreceptor cells was observed, mainly in the superior peripheral retina of both *Crb2^ΔRods^* and *Crb1^KO^Crb2^ΔRods^* mice (**B,C,E,F**). The characteristic *Crb1^KO^* phenotype, with photoreceptor dysplasia specifically in the inferior temporal retina, was also frequently observed in the inferior retina of *Crb1^KO^Crb2^ΔRods^* (**C**,**F**; arrowheads). Spidergram depicting the quantification of number of photoreceptor cell nuclei in a row in *Crb2^flox^*^/*flox*^ (grey), *Crb1^KO^Crb2^flox^*^/*flox*^ (black), *Crb2^ΔRods^* (blue), and *Crb1^KO^Crb2^ΔRods^* (red) at 6M (**G**), 9M (**H**), and 12M (**I**). A reduction in the number of photoreceptor nuclei in *Crb1^KO^Crb2^ΔRods^* retinas were observed from 6M in the superior peripheral retina (**G**). At 9M and 12M, the photoreceptor number further decreases in the superior central retina in both knockout lines (**H,I**). At 12M, the inferior retina also had a reduced number of photoreceptors (**I**). Data are presented as mean ± SEM; *n* = 3–4 mice, per genotype/time point. Note: GCL = ganglion cell layer; INL = inner nuclear layer; ONL = outer nuclear layer; RPE = retina pigment epithelium. Scale bars: 500 µm; insets: 50 µm. * *p* < 0.05, ** *p* < 0.01, *** *p* < 0.001.

**Figure 6 ijms-20-04069-f006:**
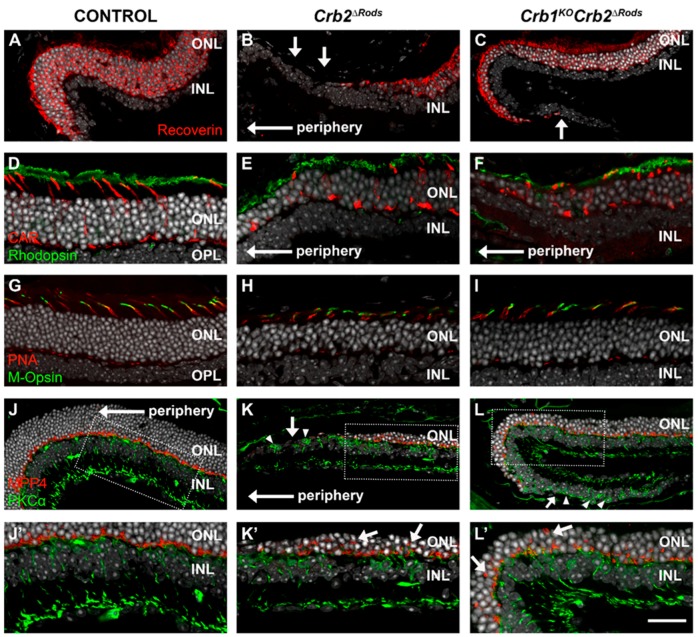
*Crb2^ΔRods^* and *Crb1^KO^Crb2^ΔRods^* mice show loss of rod photoreceptor cells. Immunohistochemistry pictures from control (**A**,**D**,**G**,**J**), *Crb2^ΔRods^* (**B**,**E**,**H**,**K**), and *Crb1^KO^Crb2^ΔRods^* (**C**,**F**,**I**,**L**) 9 month-old retinal sections stained for: recoverin (**A–C**), rhodopsin and cone arrestin (CAR) (**D–F**), peanut agglutinin (PNA) and M-opsin (**G–I**), MPP4 and PKCα (**J**–**L**, **J’**–**L’**). At 9-months-of-age, the number of photoreceptor cells marked by recoverin was reduced in the *Crb2^ΔRods^* (**C**) and *Crb1^KO^Crb2^ΔRods^* (**D**) compared to the control (**A**). Although the number of photoreceptors was reduced, in both mutant lines, rhodopsin was properly located in the outer-segments from rod photoreceptors (**E,F**). Cone photoreceptors stained with cone arresting (CAR), and with peanut agglutinin (PNA, outer segments and pedicles) and M-opsin were present in the mutant retinas and showed normal morphology (**E,F,H,I**; respectively). Photoreceptor synapses stained with MPP4 were lost and ectopically located in *Crb2^ΔRods^* (**K’**; arrows) and *Crb1^KO^Crb2^ΔRods^* (**L’**; arrows); dendritic arborization of PKCα-positive cells was also affected in the mutant retinas (**K,L**; arrowheads). Note: INL = inner nuclear layer; ONL = outer nuclear layer; OPL = outer plexiform layer. Scale bars: (**A**–**C**) 50 µm; (**D**–**L**) 25 µm.

**Figure 7 ijms-20-04069-f007:**
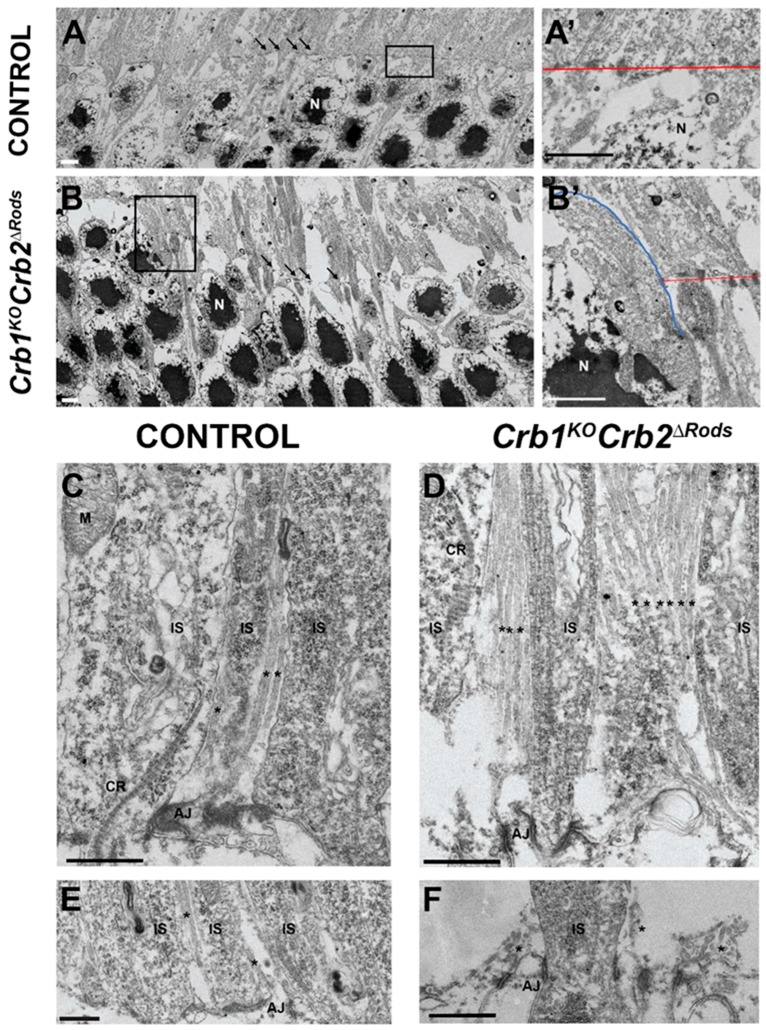
*Crb1^KO^Crb2^ΔRods^* retinas present disruptions in adherens junctions and MGC apical villi. Transmission electron microscopy pictures from control (*Crb2^flox/flox^* (**A**,**C**,**E**) and *Crb1^KO^Crb2^ΔRods^* (**B**,**D**,**F**)) in 8-month-old retinal sections. The adherens junctions were present at the outer limiting membrane in the control (**A,A’**; arrows, red line), but they were disorganized in the *Crb1^KO^Crb2^ΔRods^* retina (**B,B’**; arrows, red line). Photoreceptor cell nuclei protruding from the inner-/outer-segment layers (**B’**; blue line). In the central retina, an increased number of visible MGC apical villi were detected in *Crb1^KO^Crb2^ΔRods^* compared to control retina (**C,D**; asterisk). In the peripheral retina, MGC apical villi were normally organized in the control retina (**E**; asterisk), while in *Crb1^KO^Crb2^ΔRods^*, the MGC apical villi were collapsed (**F**; asterisk). Note: AJ = adherens junctions; CR = ciliary rootlets; IS = inner segments; N = nuclei; M = mitochondria; MGC = Müller glial cell. Scale bars: (**A**–**F**) 1 µm.

**Figure 8 ijms-20-04069-f008:**
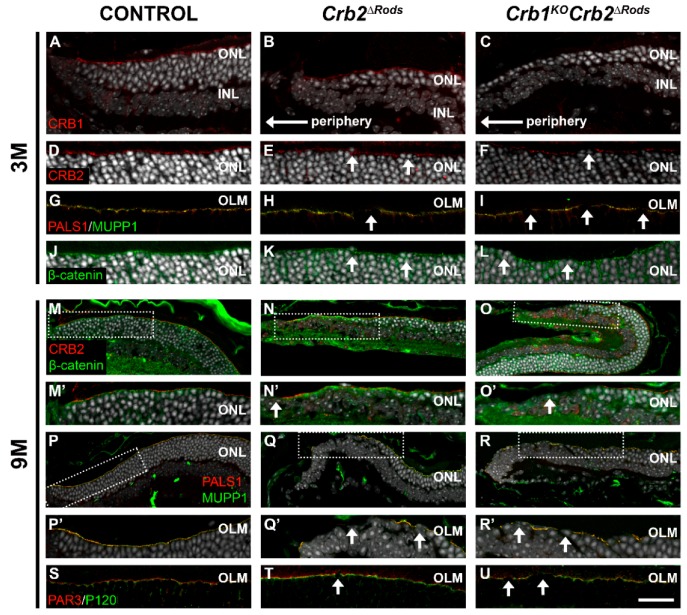
Specific ablation of Crb2 from rod photoreceptors lead to disruptions at the subapical region and at the adherens junctions. Immunohistochemistry pictures from control (*Crb2^flox^*^/flox^) (**A**,**D**,**G**,**J**,**M**,**P**,**S**), *Crb2^ΔRods^* (**B**,**E**,**H**,**K**,**N**,**Q**,**T**), and *Crb1^KO^Crb2^ΔRods^* (**C**,**F**,**I**,**L**,**O**,**R**,**U**) at different time points: 3-months-of-age (**A**–**L**) and 9-months-of-age (**M**–**U**). Retinal sections were stained for: CRB1 (**A**–**C**), CRB2 (**D**–**F**), PALS1 and MUPP1 (**G**–**I**), β-catenin (**J**–**L**), CRB2 and β-catenin (**M**–**O**), PALS1 and MUPP1 (**P**–**R**), and PAR3 and P120-catenin (**J**–**L**). At 3-months-of-age, CRB1 protein was present at the subapical region above the adherens junctions in the control (**A**) and *Crb2^ΔRods^* (**B**), however was absent in *Crb1^KO^Crb2^ΔRods^* (**C**). CRB2 is still detected at the subapical region of *Crb2^ΔRods^* (**E**) and *Crb1^KO^Crb2^ΔRods^* (**F**), due to the presence of CRB2 protein in MGCs and cone photoreceptor cells. However, disruptions of the subapical region were detected (**E,F**; arrows), mainly in the superior central retina. PALS1 and MUPP1 (**H,I**; arrows) were also lost at sites of disruption. Staining using the adherens junctions marker β-catenin showed disruption of the adherens junctions (**K,L**; arrows). Nine-month-old knockout retinas showed disruption of subapical region marker and of the adherens junctions at the superior peripheral retina (**N,N’,O,O’,Q,Q’,R,R’**) and superior central retina (**T,U**). Note: OLM = outer limiting membrane; INL = inner nuclear layer; ONL = outer nuclear layer. Scale bars: (**A**–**C, M**–**O, P**–**R**) 50 µm; (**D**–**L, M’**–**O’, P’**–**R’, S**–**U**) 25 µm.

**Figure 9 ijms-20-04069-f009:**
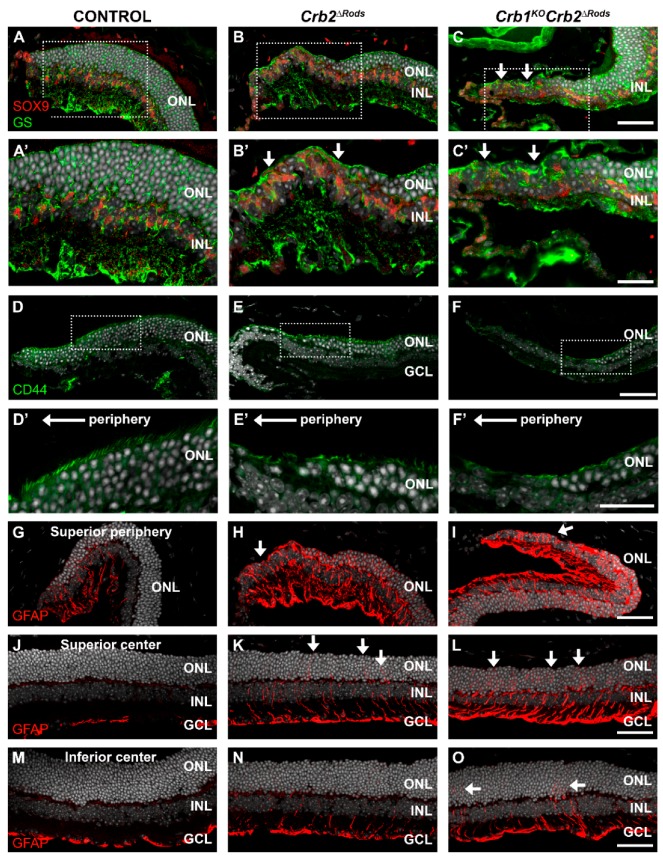
Loss of *Crb2* from rod photoreceptors leads to gliosis mainly in the peripheral superior retina. Immunohistochemistry pictures from 9-month-old control (*Crb2^flox^*^/flox^) (**A**,**D**,**G**,**J**,**M**), *Crb2^ΔRods^* (**B**,**E**,**H**,**K**,**N**), and *Crb1^KO^Crb2^ΔRods^* (**C**,**F**,**I**,**L**,**O**) retina. Retinal sections were stained with antibodies against: SOX9 and glutamine synthetase (GS) (**A**–**C**), CD44 (**D**–**F**), and glial fibrillary acidic protein (GFAP) (**G**–**O**). Müller glial cells stained with GS showed the expected radial alignment (**A,A’**). Müller glial cell morphology was affected mainly in the superior peripheral retina of the mutant retinas (**B,B’,C,C’**; arrows). CD44-positive microvilli of Müller glial cells were displaced in the *Crb2^ΔRods^* and *Crb1^KO^Crb2^ΔRods^* peripheral superior retinas (**E,E’,F,F’**), specifically where photoreceptors were lost. *Crb2^ΔRods^* and *Crb1^KO^Crb2^ΔRods^* retinas showed activated Müller glial cells, detected by an increased level of GFAP, mainly in the peripheral superior retina (H,I) and in the central superior retina (**K,L**), with a moderately increased level of GFAP in the inferior central retina (**N,O**). Note: GCL = ganglion cell layer; INL = inner nuclear layer; ONL = outer nuclear layer. Scale bars: (**A**–**C**, **D**–**F**, **G**–**O**) 50 µm; (**A’**–**C’, D’**–**F**) 25 µm.

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
