# Peer review of "CRB2 Loss in Rod Photoreceptors Is Associated with Progressive Loss of Retinal Contrast Sensitivity"

_ijms, 2019, doi:10.3390/ijms20174069_

Round 1
Reviewer 1 Report
This work clearly demonstrates that specific Cbr2 ablation in rod photoreceptors induces a disruption of retinal organization, modifies ERG and visual function. These negative effects are exacerbated with concomitant Crb1 depletion in KO animals. These results show that rod photoreceptor physiological integrity is necessary to prevent the progression of degenerative diseases such as retinitis pigmentosa.
This manuscript is well written and organized. Figures have high quality, great histology and IHCs are beautiful. However, results include many interpretative paragraphs that should be moved to discussion (e.g. L120-124, L162 and L174-179). I suggest to keep only data and figure descriptions in the results section.
Crb1KO+Crb2-flox/flox should be included in time-course graphs (Fig 1 C), Is there a reason to exclude it from this analysis?
Cbr2-Rod group should be displayed in the figure 2, even if there is no effect in special frequency, this is important data. I strongly suggest to move Supplementary Figure 3B into figure 2. Including a Crb2-flox/flox would be also useful to be included in Figure 2A, since there is no negative control.
Authors did not find any effect on pupil light reflex, however, this experiment or at least the results included in the figure are lacking positive and negative controls. Therefore, it is impossible to assume full credibility in these results.
Authors report a progressive loss of “photoreceptors” at 6, 9 and 12M (figure 5). This loss was present only in the Crb2-rod and Crb1KO+Crb2-rod, but not in Crb1KO; however, in the introduction/discussion there are references about the negative effect in Crb1KO retinas, why in this case photoreceptor loss is not induced by Crb1 absence?
There are independent IHCs for rhodopsin and Crb2, but there is no colocalization of both, it will be very useful to include a Crb2+Rhodopsin to validate that Crb2 is not expressed in the rod photoreceptors anymore. Cbr2 could be co-localized with any other Rod specific marker to validate its ablation.
Authors should speculate about the cellular mechanisms and interactions that lead to the loss of both photoreceptors and changes in the tissue organization, and the activation of reactive gliosis. Particularly, some speculation could be included in relation to the roles of every cell type during damage progression. A model figure that integrates past knowledge in this field together with this novel data will be useful for readers.
Did authors consider to include wild type (WT) animals as a control?
Reviewer 2 Report
The paper 'CRB2 loss in rod photoreceptors is associated with 2 progressive loss of retinal contrast sensitivity' by Henrique Alves et al. explores the effect of CRB2 knock out with or without the loss of CRB1 in the mouse retina. The researchers found that CRB2 in rod photoreceptor cells is required to maintain the rod photoreceptor layer and retinal electrical responses. The loss of CRB2 from rod photoreceptors leads to retinitis pigmentosa and the concomitant loss of CRB1 worsens the effect of CRB2 removal with increased visual function impairment.
The paper is very interesting and well written. I support publication of the article in its present form.
Author Response
We thank the reviewer for the positive comments.
Reviewer 3 Report
The authors investigated the CRB2 loss in rod photoreceptors is associated with progressive loss of retinal contrast sensitivity via KO mice model. This is a well design study and provided clear results. However, there are some problems had to be detail clarified
Extensive editing of English language is necessary. The article section had some problem. There was no formal hypothesis enunciated prior to the study. It is necessary to report also all the new studies of the literature on this topic which are now missing. The mechanism of pathophysiological links between iron and metabolic derangements is reported in some previous studies. We suggested authors to report current results in article.Overall, this is an interesting study. The authors provided clear information and study results.
